# Effect of Lard or Plus Soybean Oil on Markers of Liver Function in Healthy Subjects: A Randomized Controlled-Feeding Trial

**DOI:** 10.3390/foods12091894

**Published:** 2023-05-04

**Authors:** Zhiyuan Liu, Jihong Yuan, Ping Wen, Xiaofei Guo, Kelei Li, Yinpeng Wang, Ruirui Liu, Yanjun Guo, Duo Li

**Affiliations:** 1Institute of Nutrition & Health, Qingdao University, Qingdao 266071, China; lzy198822@qdu.edu.cn (Z.L.); guoxf@qdu.edu.cn (X.G.); likelei@qdu.edu.cn (K.L.); wyp18753273237@163.com (Y.W.); liuruirui20010224@163.com (R.L.); gyjforeverh@163.com (Y.G.); 2No. 2 Department of Nutrition, Chinese People’s Liberation Army General Hospital, Beijing 100853, China; jihongyuan301@126.com; 3Supply Department, Chinese People’s Liberation Army General Hospital, Beijing 100853, China; pingwen301@163.com; 4Department of Food Science and Nutrition, Zhejiang University, Hangzhou 310058, China; 5Department of Nutrition, Dietetics and Food, Monash University, Melbourne 3800, Australia

**Keywords:** lard, soybean oil, liver function, saturated fatty acid, n-6 polyunsaturated fatty acid

## Abstract

Humans have consumed lard for thousands of years, but in recent decades, it has become much less popular because it is regarded as saturated fat. Animal studies showed that lard plus soybean oil (blend oil) was more advantageous for liver health than using either oil alone. This study aims to assess the effects of blend oil on liver function markers in healthy subjects. The 345 healthy subjects were randomized into 3 isoenergetic diet groups with different edible oils (30 g/day) (soybean oil, lard, and blend oil (50% lard and 50% soybean oil)) for 12 weeks. The reductions in both aspartate aminotransferase (AST) and alanine aminotransferase (ALT) were greater in the blend oil group than in the two other groups (*p* = 0.001 and <0.001 for the interaction between diet group and time, respectively). The reductions in AST and ALT in the blend oil group were more significant compared with those in the soybean oil group (*p* < 0.001) or lard group (*p* < 0.001). There were no significant differences in the other liver function markers between the groups. Thus, blend oil was beneficial for liver function markers such as AST and ALT compared with soybean oil and lard alone, which might help prevent non-alcoholic fatty liver disease in the healthy population.

## 1. Introduction

The consumption of vegetable oils in the USA has increased by more than 1000-fold since 1909 [1], accompanied by a sharp increase in liver diseases including non-alcoholic fatty liver disease (NAFLD) [2]. Currently, n-6 polyunsaturated fatty acid (PUFA)-rich vegetable oils such as soybean oil are the most widely consumed edible oils worldwide, but in addition to their beneficial effect on blood cholesterol than saturated fat, the data on other diseases and their risk factors are very limited [3].

Soybean oil contains about 53% linoleic acid (LA) [4], an essential n-6 PUFA [5]. The excessive intake of LA may also cause some unbeneficial effects. The newly oxidized LA hypothesis suggested that LA, the most abundant fatty acid in human low-density lipoprotein (LDL), could initiate the oxidation of LDL by oxidizing itself within the LDL particles and become the most common oxidized fatty acid in LDL [6,7]. The increase in oxLDL is associated with an increased incidence of nonalcoholic steatohepatitis (NASH) in humans [8], and hepatocellular injury in experimental cholestasis and fibrosis [9]. The short-term administration of oxLDL to high-fat diet-fed mice aggravated hepatic steatosis and resulted in inflammatory cell infiltration and subsequent severe hepatic injury [10]. Meanwhile, the abundant LA in soybean oil could generate a large amount of hydroxynonenal (HNE), a major α,β-unsaturated aldehyde product of n-6 PUFA oxidation [3,11]. HNE was shown to be involved in a great number of pathologies such as metabolic diseases, neurodegenerative diseases, and cancers by modulating numerous cell processes, including oxidative stress, cell proliferation, and cell death, contributing to the occurrence of lifestyle-related diseases [11]. Thus, a moderate intake of soybean oil is essential.

Lard, an edible oil, has been used in Asian countries for centuries [12]. Because saturated fatty acids (SFAs) increase low-density lipoprotein cholesterol (LDL-C), the consumption of lard has significantly reduced in recent years. However, the majority of earlier lard-related studies utilized a high-fat diet [13,14,15], which is very dissimilar from the diet of Asian nations, including China. While moderate lard consumption was shown in human intervention trials to increase LDL-C in comparison to plant oils such as soybean oil, olive oil, and palm olein [16,17], in most individuals, the increase in LDL-C is not due to an increase in the levels of small dense LDL particles, but rather, larger LDL particles, which are much less strongly associated with CVD risk [18]. Additionally, it was reported that brown sauce braised pork is a popular dish among Chinese people with longevous lifespans [19]. Lard has a fairly balanced fatty acid composition (52.1% SFAs, 35.8% monounsaturated fatty acids (MUFAs), and 11.6% PUFAs) [4], which may make it more likely to satisfy human nutritional needs. Accordingly, further investigation is needed to elucidate the effect of moderate lard intake on human health.

Recently, several animal studies demonstrated that lard plus soybean oil alleviated NAFLD compared with soybean oil partly by reducing the AST and ALT [4] and had a remarkable anti-obesity effect compared with lard and soybean oil alone [19]. We believe that lard plus soybean oil may have a beneficial effect on liver function in humans, but no related clinical trials exist. Thus, the present study aims to investigate the effects of lard, lard plus soybean oil, and soybean oil on markers of liver function in healthy subjects.

## 2. Materials and Methods

### 2.1. Subject Enrollment

This study was a 12-week randomized controlled-feeding trial using a parallel 3-arm design. Participants were subjects who worked and lived at People’s Liberation Army General Hospital, implying that all participants consumed the same cooking oil (soybean oil) prior to the current study; thus, no run-in period was included. The inclusion criteria were as follows: (1) between 18 and 65 years old; (2) body mass index (BMI) (calculated as weight in kilograms divided by height in meters squared) between 16 and 40. The exclusion criteria were as follows: (1) intolerance or allergy to lard; (2) participation in other studies involving drug or dietary intervention in the past 3 months; (3) unable to comply with our protocol; (4) reported history of severe diabetes or CVD or cancer; (5) being pregnant, or planning pregnancy; (6) particular nutritional habits such as vegetarianism. 

### 2.2. Randomization of Participants and Intervention

All the included participants were randomly allocated to one of the three groups using computer-generated random numbers as follows: soybean oil group (n = 116), blend oil group (n = 116), and lard group (n = 113). Measurements were carried out by clinical and laboratory staff who were unaware of group assignment. Even though they were aware of the diet assignment, the meal providers were excluded from the rest of the study, including the measures and data analysis that followed. It was challenging to blind participants because lard becomes solid when it is cold, though they were not informed of the allocated diet.

In accordance with the Dietary Guidelines for Chinese Residents, the daily amount of edible oil in each subject was restricted to 30 g [19], which was added to items during cooking. With the exception of the type of cooking oil, each participant in the three groups consumed the same meals, which were provided by the hospital canteen, suggesting that they shared equal energy, the same macronutrient ratios (approximately 55% of energy from carbohydrates, 30% from fat, and 15% from protein, based on the current macronutrient intake in China), and the same cooking methods (Table 1). Cooking oil fatty acid compositions were analyzed and are shown in Table 2. All 3 meals were given to participants without charge each day for the duration of the 12-week intervention. 

For each diet, four 7-day cycling recipes (each for three weeks periodically) were created. During breakfast, participants were allowed to consume non-oily foods as much as they needed. In general, participants were required to eat their lunch and dinner (packaged) on-site each day while being supervised by staff. All participants were advised to finish the food supplied, and refrain from eating any other high-fat items. Compliance was assessed throughout the time that meals were served on-site through direct observation of the number of missed meals, the amount of food that was left uneaten, and the subjective assessment of the clinic personnel. All participants were advised to maintain their previous snacking and exercise habits.

### 2.3. Measurements of Anthropometric and Biochemical Parameters

At baseline, weeks 4, 8, and 12, all outcome measurements—including body weight, waist circumference, markers of liver function, lipid profiles, blood pressure, and glucose—were obtained. Overnight fasting (12 to 14 h) blood samples were collected at baseline and every 4 weeks during the study. A Hitachi 3110 blood biochemistry analyzer (Tokyo, Japan) was used to measure biochemical parameters in the sera. Waist circumference and body weight were recorded with participants wearing light clothing and no shoes. Certified medical professionals used a standardized mercury sphygmomanometer to record the participants’ resting blood pressure after 5 min of rest in the participants’ canteen or rest area. 

### 2.4. Dietary Intake Assessment

Data on food intake were gathered using a three-day dietary record (two weekdays and one weekend day) at baseline and a seven-consecutive-day menu throughout the intervention period. The Nutrition System of Traditional Chinese Medicine Combining with Western Medicine, version 11.0, created by the Medical College, Qingdao University, Qingdao, Shandong, China, was then used to analyze the data.

### 2.5. Measurements of Erythrocyte Fatty Acids

Erythrocyte phospholipid fatty acids were measured using gas chromatography as previously described [20]. Briefly, total lipids were extracted from erythrocytes with methanol/chloroform (*v*/*v*, 1:1), and phospholipid fractions were separated from the total lipids using thin-layer chromatography. Then, the phospholipid fatty acids were converted into methylesters, which were subsequently extracted into n-hexane and dehydrated via anhydrous Na_2_SO_4_. Finally, fatty acid methylesters were filtered using Sep-Pak Silica column (Alltech Associates Inc., Deerfield, IL, USA) before gas chromatography separation and analysis.

### 2.6. Sample Size and Power Analysis

The sample size was determined based on a prior human intervention trial adopting meals enriched with SFAs or PUFAs [21]. The current study was designed to detect a difference in ALT of 3 U/L following a 12-week intervention. We predicted a standard deviation (SD) of 7 U/L and a dropout rate of 20%. Based on these presumptions, it was determined that 112 subjects in each group would have a power of 90% for the 2-sided test with a significance level of 0.05. 

### 2.7. Statistical Analyses

The intention-to-treat (ITT) methodology was used for the main analyses. Given the skewed distributions, we used generalized linear mixed model (GLMM) to examine the longitudinal changes in serum biochemical markers, physiological parameters, and blood pressure over time among three intervention groups. Additionally, GLMM is an extension of the flexible linear mixed model that also incorporates random effects, which are helpful for accommodating the heterogeneity found in repeated measures (longitudinal design) [22]. Considering that a mixed-model analysis without any ad hoc imputation for missing data would offer equal or higher power than a dose analysis with data imputation, multiple imputations of missing data were not utilized [23]. Time, group, and time × group interaction were the fixed effects. Participant identifier was regarded as a random effect.

The *p* value for time × group interaction provided a test whether markers of liver function among the three groups differed significantly during the course of the investigation. A parallel approach was used for other results, including waist circumference, body weight, lipid profiles, blood glucose, and blood pressure. Analyses for AST and ALT were complemented by a per-protocol method in addition to ITT analysis, while analyses of other outcomes were simply conducted using ITT principles. SPSS version 26 (IBM, Armonk, NY, USA) was used for all statistical analyses. 

### 2.8. Study Ethics

The trial was registered at the Chinese Clinical Trial Registry (ChiCTR) Platform (ChiCTR2100051021, https://www.chictr.org.cn/showprojEN.html?proj=133195 (10 September 2021), date of registration: 10 September 2021). The trial was approved by the Ethics Committee of the Medical College of Qingdao University (QDU-HEC-2021105), China. All subjects signed a consent form before participating.

## 3. Results

### 3.1. Study Population

The 345 healthy subjects (mean age 33.5 years, 59% women) from the People’s Liberation Army General Hospital (Beijing, China) were randomized to 3 isoenergetic diets that provided 3 different edible oils (30 g/day) (soybean oil group, n = 116; lard plus soybean oil group (50% lard and 50% soybean oil, blend oil group), n = 116; lard group, n = 113) for 12 weeks. A total of 253 (73.3%) subjects completed the study. The percentages of male and female subjects who completed the study were 72.0% and 74.3%, respectively. The retention rates at 12 weeks were 80.2%, 69.8%, and 69.9% in the soybean oil group, blend oil group, and lard group, respectively (Figure 1). The baseline characteristics of the participants are shown in Table 3. The nutritional compositions of the diets during the baseline and intervention periods among the three groups are shown in Table 1. All 345 participants were included in the ITT analysis. 

### 3.2. Changes in AST and ALT

After 12 weeks of intervention, the ITT analysis showed that the reductions in both AST and ALT were greater in the blend oil group than in the two other groups (*p* = 0.001 and <0.001 for the interaction between the diet group and time, respectively) (Figure 2). The reductions in AST and ALT in the blend oil group were more significant compared with those in the soybean oil group with *p* < 0.001 and *p* < 0.001, and the lard group with *p* < 0.001 and *p* < 0.001, respectively (Figure 2). The AST changes at 12 weeks were −2.0 U/L (95% CI −3.2 to −0.8) in the blend oil group, −0.3 U/L (95% CI −1.2 to 0.6) in the soybean oil group, and 1.0 U/L (95% CI −0.6 to 2.6) in the lard group. The corresponding ALT changes were −4.5 U/L (95% CI −7.3 to −1.7), −0.1 U/L (95% CI −2.1 to 1.8), and 1.8 U/L (95% CI −0.5 to 4.0), respectively (Table 4). 

Among the 253 participants who completed the intervention, greater AST and ALT reductions were also observed in the blend oil group relative to the two other groups (*p* = 0.002 and *p* < 0.001 for the interaction between diet group and time, respectively) (Table 4). Additionally, the reductions in AST and ALT in the blend oil group were still more significant compared with those in the soybean oil group with *p* = 0.001 and *p* < 0.001, and the lard group with *p* < 0.001 and *p* < 0.001, respectively. 

### 3.3. Subgroup Analyses of AST and ALT

Both age and gender remarkably influenced the effects of different oils on AST (Table 5). More specifically, the participants who were <45 years old and male had significant decreases in AST in the blend oil group compared with the two other groups (*p* = 0.001 and 0.001 for the interactions between diet group and time, respectively); while the reductions in AST in the participants who were ≥45 years old and female were not significant (*p* = 0.398 and 0.076, respectively) (Table 5). Meanwhile, age and gender did not influence the effects of different oils on ALT. More specifically, both participants <45 years old and ≥45 years old, male and female, had significant decreases in ALT in the blend oil group compared with the two other groups (Table 5). 

### 3.4. Other Biochemical Markers and Physiological Parameters

There were no significant differences among the three groups in other markers of liver function, including alkaline phosphatase (ALP), gamma-glutamyltransferase (γ-GT), total protein (TP), albumin (ALB), globulin (GLB), albumin/globulin (A/G), total bilirubin (T-BIL), direct bilirubin (D-BIL), and indirect bilirubin (I-BIL) (Table 6). However, remarkable reductions in both SBP and DBP were observed in the blend oil group than in the two other groups (*p* = 0.029 and 0.003 for the interaction between diet group and time, respectively) (Table 7). No significant differences were observed between the diet groups for any other biochemical markers (such as triglyceride (TG), total cholesterol (TC), high-density lipoprotein cholesterol (HDL-C), TC/HDL-C, LDL-C, free fatty acid (FFA), or glucose) or physiological parameters (including waist circumference and body weight) (Table 7). 

### 3.5. Fatty Acid Composition of Erythrocyte Membrane

Table 8 compares the amounts of erythrocyte membrane fatty acids in the three groups among subjects who donated blood at both the baseline and 8-week points. When compared to the two other groups at the end of the 8-week intervention, the soybean oil group’s C18:3n-3 levels were significantly higher (*p* = 0.001 for the interaction between diet group and time). No difference was observed in the other thirteen main fatty acids. 

## 4. Discussion

To the best of our knowledge, the present study is the first randomized controlled-feeding trial to explore the effect of isocaloric diets enriched with lard or lard plus soybean oil on markers of liver function in healthy subjects. The results showed that the reductions in both AST and ALT were significantly greater in the blend oil group than in the two other groups throughout the intervention (Figure 2). Meanwhile, greater reductions in AST and ALT in the blend oil group were observed in comparison to those in the soybean oil group or lard group (Figure 2). However, no significant differences were observed among the three groups in other markers of liver function, including ALP, γ-GT, TP, ALB, GLB, A/G, T-BIL, D-BIL, and I-BIL. Our findings indicate that blend oil (lard plus soybean oil) may have an effect on improving liver function in healthy subjects compared with soybean oil and lard alone. 

ALT is primarily localized to the liver, but AST is present in a wide variety of tissues such as the heart, skeletal muscle, kidney, brain, and liver [24]. Both AST and ALT levels are increased to some extent in almost all liver diseases due to hepatocellular necrosis, including NAFLD, cirrhosis, non-alcoholic steatohepatitis (NASH), etc. [24]. In the present study, the significant reductions in both AST and ALT in the blend oil group suggest that blend oil is likely to decrease hepatocellular necrosis, which might be beneficial to prevent liver diseases, including NAFLD, which affects approximately 30% of the population [25].

Currently, clinical trials related to the effects of lard or lard plus soybean oil on markers of liver function are very scarce; this gap is now filled by the present study. Our significant decreases in AST and ALT in the blend oil group relative to the soybean oil group are identical to those of a recent animal study that evaluated the effects of lard, soybean oil, or their blend oil (46% lard and 54% soybean oil) on NAFLD by following a low-fat, high-carbohydrate diet [4]. However, our significant decreases in AST and ALT in the blend oil group relative to the lard group are not identical to those of that animal study [4], which were not significant. The possible explanation is that in the present study, lard may have been overconsumed because it accounted for 13.2% of the total energy, more than the 10% recommended for SFA fat by dietary guidelines [26]. In line with this, an animal study also reported that the overconsumption of lard, which contained 30% lard in the rodent chow, led to striking increases in AST and ALT [15]. The amount of lard (14.6% of total energy) used in the above-mentioned animal study is likely to be appropriate for mice, as shown by the results of biochemical markers and tissue sections [4]. The other possible reasons include different intervention periods and species differences.

The significantly greater reductions in AST and ALT following the blend oil compared with the two other oils may have been due to its relatively balanced fatty acid ratio (SFA: MUFA: PUFA = 0.5:0.7:1.0 for blend oil; 2.3:2.8:1.0 for lard; 0.2:0.3:1.0 for soybean oil) (Table 2). Given that palmitic acid (PA), oleic acid (OA), and linoleic acid (LA) are the major fatty acids in the three oils and represent the main SFA, MUFA, and PUFA, respectively (Table 2), the following discussion will focus on these three fatty acids (Figure 3). AST and ALT derive from damaged hepatic cells into the blood after their death [24], and apoptosis is the main reported fatty acid-related cell death [27,28,29]; therefore the discussion below will be around the relationship between fatty acids and apoptosis-induced or simply cell death-induced increases in AST and ALT (Figure 3).

Several molecular mechanisms are reported to be involved in PA-associated apoptosis, including endoplasmic reticulum (ER) stress, reactive oxygen species (ROS) generation, and ceramide production (Figure 3) [30,31]. (i) ER stress: ER stress has been postulated as a major initiator of lipid-induced toxicity [32]. In human and mouse hepatocytes, PA at a lipotoxic concentration triggered the early activation of ER stress-related kinase ribosomal protein S6 kinase 1 (S6K1), induced the apoptotic transcription factor CHOP, activated caspase 3, and increased the percentage of apoptotic cells [33]. (ii) Ceramide production: In PA-mediated HepG2 (a human liver cancer cell line) apoptosis, PA-activated dihydroceramide desaturase 1 (DES1) → ceramide → caspase 9 → caspase 3 signaling [27]. (iii) ROS generation: PA induced ROS accumulation in H4IIEC3 rat hepatocytes, which preceded the onset of apoptosis as indicated by the induction of caspases 3/7 [28]. However, it is worth noting that whereas high-level PA induced lipotoxic effects, low-level PA benefited liver cells by boosting mitochondrial metabolism and compromised both high-level-PA-induced lipotoxicity and CCl4-generated hepatotoxicity [34]. 

In contrast to PA, OA elicits several beneficial effects. For example, OA suppressed the toxic effects of PA on ER stress activation and lipoapoptosis by inhibiting the PA-induced activation of S6K1 in hepatocytes [33]. Moreover, OA attenuated PA-induced apoptosis in the HepG2 cells [29]. The possible mechanism is OA-induced autophagy via the Beclin 1/class-III PI3 kinase complex, contributing to the suppression of PA-induced apoptosis [29]. 

LA is indispensable for human health as an essential fatty acid [5]. The addition of LA to the PA-treated HepG2 cells prevented apoptotic transcription factor CHOP induction [35]. However, the excessive consumption of LA, a kind of n-6 PUFAs and a main fatty acid in soybean oil (≈58.5%, Table 2), may also cause some unbeneficial effects. The ‘calpain–cathepsin hypothesis’ explains the molecular cascade from LA to cell death (Figure 3) [3]. LA generates oxidative stress-induced hydroxynonenal (HNE), which is one of the most abundant and toxic aldehydes generated through ROS-mediated peroxidation of lipids [36] and is considered to play a crucial role in the oxidative injury of biomolecules [37]. HNE results in calpain activation induced by Ca^2+^ mobilization via GPR109A and the subsequent carbonylation of Hsp70.1 (also called Hsp70 or Hsp72) [3,38,39]. The activated calpain cleaves carbonylated Hsp70.1, contributing to lysosomal rupture which induces cathepsin leakage and subsequent cell death, and autophagy failure which leads to the formation of microcysts or vacuoles containing garbage proteins or membrane lipid-degradation components and cell death. 

Based on the above-mentioned and emerging evidence, several reasons may explain why blend oil is superior to lard to ameliorate markers of liver function. First, blend oil has lower PA levels (16.3% in blend oil versus 25.1% in lard; Table 2), while the PA intake in the lard group may be excessive. In the lard group, lard accounted for 13.2% of the total energy (6.6% in the blend oil group), which exceeded the 10% recommended for SFA fat by dietary guidelines [26]. The excessive consumption of PA may cause higher levels of apoptosis in liver cells, resulting in higher AST and ALT in serum. Second, blend oil has higher LA levels (38.7% in blend oil versus 14.8% in lard; Table 2), while the LA intake in the lard group may be deficient. The Food and Agriculture Organization suggests 2.5–9.0% dietary energy for LA [40]. In the present study, the LA intakes were about 1.9% and 5.1% in the lard and blend oil groups, respectively. LA is an essential fatty acid and can suppress CHOP-induced apoptosis [35], which might explain why blend oil is better than lard for liver function. Third, although blend oil has lower OA (31.2% in blend oil versus 42.4% in lard; Table 2), there is also less PA in it (16.3% in blend oil versus 25.1% in lard; Table 2).

Meanwhile, a few reasons possibly explain why blend oil is better than soybean oil to improve markers of liver function. While the higher LA levels in the soybean oil (58.5% in soybean oil versus 38.7% in blend oil; Table 2) may suppress more CHOP-induced apoptosis, it is also likely to generate more HNE [36], which could contribute to higher levels of cell death [3], resulting in increases in AST and ALT in the soybean oil group. On the other hand, blend oil has higher OA (31.2% in blend oil versus 21.9% in soybean oil; Table 2), which could suppress ER stress [33] and increase autophagy [29], contributing to a lower level of apoptosis and possibly consequent decreases in AST and ALT. One might argue that the higher PA in blend oil (16.3% in blend oil versus 8.7% in soybean oil; Table 2) may lead to higher PA-induced apoptosis and subsequently higher AST and ALT. However, the PA intake in both groups is quite limited. Even after adding all the SFAs together, the SFAs in both groups have not exceeded the upper limit of 10% of the total energy recommended by dietary guidelines (Table 1) [26]. On the contrary, a low level of PA is reported to have beneficial effects on liver mitochondrial metabolism, liver protection, and combating oxidative stress [34,41]. Overall, the improvement in markers of liver function in the blend oil group relative to the lard group and soybean oil group is probably owing to its relatively balanced fatty acid ratio. 

Although moderate blend oil intake (30 g/day) may not only be hepatoprotective and harmless, but even beneficial to the cardiovascular system according to the improved blood pressure (Table 7), the effects of high blend oil intake on liver function and the cardiovascular system are still unknown. Given that high lard intake may increase the type 2 diabetes risk [42], blood pressure [43], and serum lipids [44], and high soybean oil intake may also slightly increase the type 2 diabetes risk [42], and repeatedly heated soybean oil caused an increase in blood pressure [45], it is suggested that the moderate intake of lard plus soybean oil (without repeatedly heating) may be healthier for liver function and the cardiovascular system than the use of either oil alone. 

Our trial has several limitations. Considering our relatively short-term intervention period of 12 weeks, we cannot predict what would have happened if the intervention had been kept for a longer period or whether the blend oil could be used to prevent NAFLD. Meanwhile, the present study focuses on healthy Chinese adults; therefore, we cannot figure out whether blend oil will show similar beneficial effects on other races or certain specific groups such as people with NAFLD. In addition, the current findings are only about the intermediate metabolic risk indicators, indicating that they cannot be directly extended to findings for clinical endpoints.

The strengths of this study include the study design of a randomized controlled-feeding trial, the similar results observed when analyzing the data according to intention-to-treat and per-protocol, a relatively high completion rate (73.3%), and a large sample size, all of which enhance our confidence in the findings.

## 5. Conclusions

In conclusion, our study shows that blend oil (50% lard plus 50% soybean oil) has beneficial effects on markers of liver function such as AST and ALT compared with lard and soybean oil but has no significant effects on other markers of liver function. These findings will shed new light on the impact of a lard–vegetable oil mixture on human liver health, especially given that NAFLD affects 30% of the population, and high circulating ALT and AST are widely used surrogates for NAFLD. Future studies are needed to elucidate (1) the effects of blend oil on preventing liver diseases, including NAFLD, and the mechanism behind these effects, and (2) the effects of blend oil (lard plus soybean oil) on the liver function in patients with liver diseases.

## Figures and Tables

**Figure 1 foods-12-01894-f001:**
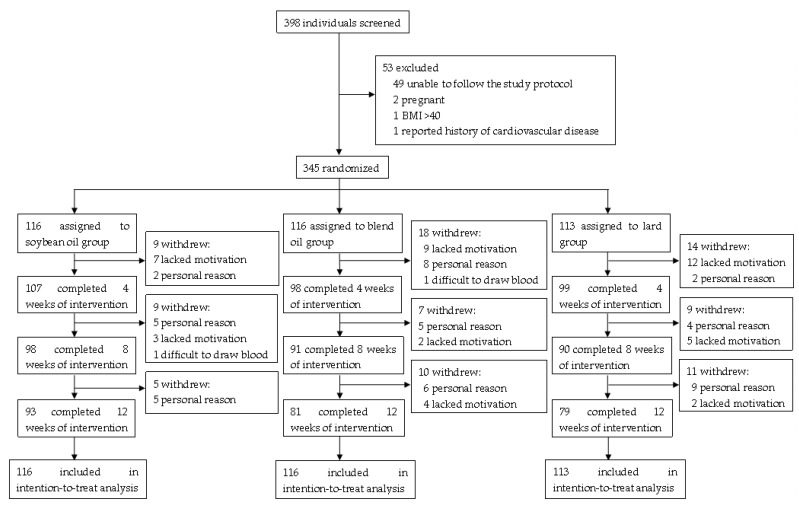
Flow chart of the present randomized controlled trial.

**Figure 2 foods-12-01894-f002:**
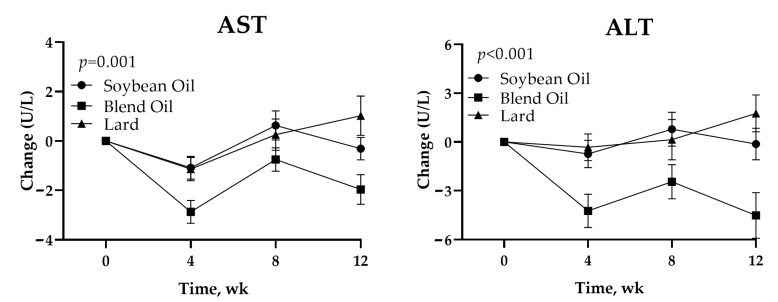
Mean changes and diet contrast in AST and ALT. Error bars indicate SEM. Data are based on a mixed-model analysis of variance. The *p* value at the upper left indicates the test of whether the change between baseline and intervention period (mean of every 4 weeks) differed significantly between participants assigned to three diet groups. The *p* values for the comparison between soybean oil group and blend oil group are <0.001 for AST and <0.001 for ALT. The *p* values for the comparison between blend oil group and lard group are <0.001 for AST and <0.001 for ALT. The *p* values for the comparison between soybean oil group and lard group are 0.502 for AST and 0.524 for ALT. AST, aspartate aminotransferase; ALT, alanine aminotransferase; wk, week.

**Figure 3 foods-12-01894-f003:**
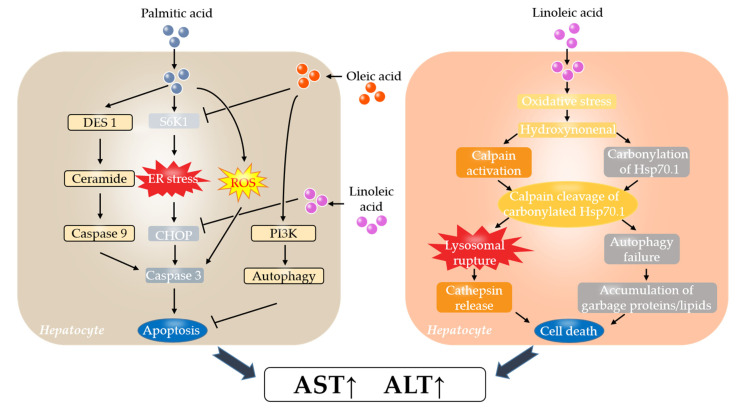
The proposed underlying mechanisms for the improvement in AST and ALT by blending lard and soybean oil. Blend oil has relatively balanced fatty acid ratio (SFAs: MUFAs: PUFAs = 0.5:0.7:1.0 for blend oil; 0.2:0.3:1.0 for soybean oil; 2.3:2.8:1.0 for lard) (Table 2). Palmitic acid (PA), oleic acid (OA), and linoleic acid (LA) are the three major fatty acids in the three oils and are also the main SFA, MUFA, and PUFA in the three oils, respectively (Table 2). The focus of this figure is on these three main fatty acids. (i) Picture on the left shows that PA induces apoptosis through ER stress, ROS generation, and ceramide production. While OA suppresses PA-induced apoptosis through inhibiting PA-induced activation of S6K1 and enhancing autophagy via PI3K, LA suppresses PA-induced apoptosis through preventing CHOP induction. (ii) Picture on the right shows the ‘calpain–cathepsin hypothesis’ explaining the molecular cascade from LA to cell death. Overconsumption of LA generates large amounts of hydroxynonenal (HNE) due to oxidative stress, resulting in carbonylation of Hsp70.1 (also called Hsp70 or Hsp72) and Ca^2+^ mobilization-induced calpain activation. The activated calpain cleaves carbonylated Hsp70.1, causing cathepsin leakage and autophagy failure and subsequent cell death. Abbreviations: ALT, alanine aminotransferase; AST, aspartate aminotransferase; DES1, dihydroceramide desaturase 1; ER stress, endoplasmic reticulum stress; ROS, reactive oxygen species; S6K1, ribosomal protein S6 kinase 1; PI3K, phosphatidylinositol 3-kinase.

**Table 1 foods-12-01894-t001:** Nutritional composition of diets during baseline and intervention among the three groups.

	Baseline			Intervention		
	Soybean Oil Group	Blend Oil Group	Lard Group	Soybean Oil Group	Blend Oil Group	Lard Group
Energy (male), kcal	2051(112.9)	2048 (112.2)	2045 (114.4)	2047 (152.7)	2034 (152.7)	2021 (152.7)
Energy (female), kcal	1661 (74.0)	1634 (80.6)	1647 (84.1)	1706 (127.2)	1695 (127.2)	1684 (127.2)
Carbohydrate (E %)	53.0 (4.4)	52.8 (5.6)	52.7 (5.1)	58.7 (3.0)	59.3 (3.1)	60.0 (3.1)
Protein (E %)	16.2 (1.8)	16.2 (2.7)	16.2 (2.6)	13.6 (1.1)	13.6 (1.1)	13.6 (1.1)
Total fat (E %)	30.5 (3.7)	30.7 (4.9)	30.7 (4.4)	27.7 (2.9)	26.9 (2.9)	26.2 (2.9)
SFA (E %)	6.5 (1.3)	6.7 (1.7)	6.7 (1.4)	5.9 (0.9)	7.5 (0.9)	9.1 (0.8)
MUFA (E %)	8.0 (1.7)	8.3 (2.2)	8.2 (1.7)	7.6 (1.1)	8.9 (1.1)	10.3 (1.1)
PUFA (E %)	11.0 (2.0)	11.4 (1.9)	11.4 (1.6)	10.0 (0.7)	6.3 (0.5)	2.7 (0.5)
S:M:P ratio	0.6:0.7:1	0.6:0.7:1	0.6:0.7:1	0.6:0.8:1	1.2:1.4:1	3.4:3.9:1
Fiber (g/day)	14.2 (2.6)	14.1 (2.8)	13.9 (3.3)	17.6 (1.9)	17.6 (1.9)	17.6 (1.9)
Cholesterol (mg/day)	324.8 (142.8)	317.3 (179.7)	330.3 (165.7)	324.1 (47.9)	343.9 (47.9)	363.7 (47.9)

Data are expressed as means ± SD for baseline nutritional composition of diets according to dietary intake data over 3 days (2 weekdays and 1 weekend day) at baseline. Additionally, data are expressed as means for intervention nutritional composition of diets according to a 7-consecutive-day menu analysis. Values were assessed using the Nutrition System of Traditional Chinese Medicine Combining with Western Medicine, version 11.0, developed by the Medical College, Qingdao University, Shandong, China. The system includes food composition data, permitting calculation of nutrient intake from a 3-day dietary record and the menus. E, energy; SFA, saturated fatty acid; MUFA, monounsaturated fatty acid; PUFA, polyunsaturated fatty acid; S, SFA; M, MUFA; P, PUFA.

**Table 2 foods-12-01894-t002:** Fatty acid composition of cooking oils.

Fatty Acid	Soybean Oil	Blend Oil	Lard
SFA (%)			
C14:0 (myristic acid)	0.06	0.68	1.49
C16:0 (palmitic acid)	8.66	16.31	25.12
C17:0 (margaric acid)	0.03	0.09	0.13
C18:0 (stearic acid)	1.76	6.34	11.58
C20:0 (arachidic acid)	0.15	0.12	0.09
MUFA (%)			
C16:1n-7 (palmitoleic acid)	0.30	1.11	2.26
C18:1n-9 (oleic acid)	21.89	31.22	42.41
C20:1n-9 (paullinic acid)	0.08	0.29	0.56
PUFA (%)			
n-6 PUFA			
C18:2n-6 (linoleic acid)	58.51	38.67	14.84
C20:2n-6 (dihomo-linoleic acid)		0.22	0.48
C20:3n-6 (dihomo-γ-linolenic acid)		0.02	0.04
C20:4n-6 (arachidonic acid)		0.08	0.18
n-3 PUFA			
C18:3n-3 (linolenic acid)	8.57	4.86	0.82
SFA (%)	10.7	23.5	38.4
MUFA (%)	22.3	32.6	45.2
PUFA (%)	67.1	43.8	16.4
S:M:P ratio	0.2:0.3:1.0	0.5:0.7:1.0	2.3:2.8:1.0
PA:OA:LA	0.1:0.4:1.0	0.4:0.8:1.0	1.7:2.9:1.0
n-6/n-3	6.8	8.0	19.1

The fatty acid profiles were identified using Aglient Gas Chromatograph 7820A. SFA, saturated fatty acid; MUFA, monounsaturated fatty acid; PUFA, polyunsaturated fatty acid; S, SFA; M, MUFA; P, PUFA; PA, palmitic acid; OA, oleic acid; LA, linoleic acid.

**Table 3 foods-12-01894-t003:** Baseline characteristics of the intention-to-treat population.

	Soybean Oil Group (n = 116)	Blend Oil Group (n = 116)	Lard Group (n = 113)	Reference Intervals
Age (years)	34.0 (12.4)	33.1 (12.3)	33.5 (12.0)	/
Female (n (%))	69 (59.5%)	61 (52.6%)	72 (63.7%)	/
Male (n (%))	47 (40.5%)	55 (47.4%)	41 (36.3%)	/
Weight (kg)	65.0 (13.6)	67.0 (13.7)	64.4 (12.2)	/
BMI (kg/m^2^)	24.0 (4.0)	24.5 (4.0)	24.2 (4.0)	18.5–23.9
Waist circumference (cm)	78.2 (11.5)	78.7 (12.3)	77.1 (11.3)	<85 (male) or <80 (female)
Blood pressure (mm Hg)				
Systolic	115.8 (12.3)	117.4 (13.8)	115.6 (9.7)	90–140
Diastolic	74.9 (8.7)	76.0 (10.2)	74.2 (8.3)	60–90
Blood biomarkers				
TC (mmol/L)	4.56 (0.74)	4.71 (0.78)	4.66 (0.81)	<5.18
LDL-C (mmol/L)	2.55 (0.58)	2.67 (0.67)	2.62 (0.62)	<3.37
TG (mmol/L)	1.12 (0.79)	1.09 (0.59)	1.13 (0.75)	<1.70
HDL-C (mmol/L)	1.68 (0.38)	1.70 (0.28)	1.69 (0.32)	>1.04
TC/HDL-C	2.81 (0.62)	2.83 (0.55)	2.81 (0.54)	/
FFA (μmol/L)	754.3 (275.2)	730.4 (280.4)	778.7 (262.5)	100–760
Glucose (mmol/L)	4.73 (0.72)	4.76 (0.74)	4.84 (1.08)	3.90–6.10
AST (U/L)	21.8 (7.8)	23.1 (6.1)	22.1 (6.6)	13–35
ALT (U/L)	20.2 (14.8)	24.0 (16.7)	22.1 (16.3)	7–40

Data are means (SD) or n (%). BMI, body mass index; TC, total cholesterol; LDL-C, low-density lipoprotein cholesterol; TG, triglyceride; HDL-C, high-density lipoprotein cholesterol; FFA, free fatty acid; AST, aspartate aminotransferase; ALT, alanine aminotransferase.

**Table 4 foods-12-01894-t004:** Intention-to-treat and per-protocol analyses of AST and ALT changes from baseline.

Variable	Week	Soybean Oil Group	Blend Oil Group	Lard Group	*p*-Value *		
					Time	Group	Time × Group
ITT (n = 345)							
AST (U/L)	0	21.8 (20.3, 23.2)	23.1 (22.0, 24.2)	22.1 (20.8, 23.3)	<0.001	<0.001	0.001
	Δ4	−1.1 (−2.0, −0.2) ^a^	−2.9 (−3.8, −2.0) ^b^	−1.1 (−2.1, −0.2) ^a^			
	Δ8	0.6 (−0.5, 1.8)	−0.7 (−1.7, 0.2)	0.3 (−1.0, 1.5)			
	Δ12	−0.3 (−1.2, 0.6)	−2.0 (−3.2, −0.8) ^b^	1.0 (−0.6, 2.6) ^a^			
ALT (U/L)	0	20.2 (17.5, 22.9)	23.9 (20.9, 27.0)	22.1 (19.0, 25.1)	0.003	<0.001	<0.001
	Δ4	−0.7 (−2.4, 0.9) ^a^	−4.2 (−6.3, −2.2) ^b^	−0.3 (−2.0, 1.3) ^a^			
	Δ8	0.8 (−1.3, 2.9) ^a^	−2.4 (−4.5, −0.4) ^b^	0.1 (−2.3, 2.6)			
	Δ12	−0.1 (−2.1, 1.8) ^a^	−4.5 (−7.3, −1.7) ^b^	1.8 (−0.5, 4.0) ^a^			
PP (n = 253)							
AST (U/L)	0	21.8 (20.1, 23.5)	23.0 (21.5, 24.4)	21.5 (20.2, 22.7)	<0.001	<0.001	0.002
	Δ4	−0.9 (−1.9, 0.1) ^a^	−2.6 (−3.6, −1.6) ^b^	−1.1 (−2.2, 0.0) ^a^			
	Δ8	0.9 (−0.3, 2.1)	−0.6 (−1.6, 0.4)	0.6 (−0.8, 2.0)			
	Δ12	−0.3 (−1.2, 0.7) ^a^	−2.0 (−3.2, −0.8) ^b^	1.0 (−0.6, 2.6) ^a^			
ALT (U/L)	0	20.2 (17.0, 23.4)	24.4 (20.3, 28.5)	20.4 (17.8, 23.0)	0.021	<0.001	<0.001
	Δ4	−0.6 (−2.5, 1.3) ^a^	−4.3 (−6.5, −2.0) ^b^	0.0 (−1.9, 1.9) ^a^			
	Δ8	0.9 (−1.2, 3.1)	−2.0 (−4.3, 0.2) ^b^	1.2 (−1.1, 3.6) ^a^			
	Δ12	−0.1 (−2.1, 1.9) ^a^	−4.5 (−7.3, −1.7) ^b^	1.7 (−0.5, 4.0) ^a^			

All values are means (95% CIs). * Data are based on mixed-model analysis of variance; changes from baseline were calculated by subtracting 4-, 8-, and 12-week data from baseline data. ITT, intention-to-treat; PP, per-protocol; AST, aspartate aminotransferase; ALT, alanine aminotransferase; Δ, change from baseline. ^a,b^ Different superscripts indicate significant differences between groups. No superscript means no difference compared with any other group.

**Table 5 foods-12-01894-t005:** Subgroup analyses of AST and ALT in the intention-to-treat participants among the three groups during the intervention period.

Variable	Week	Soybean Oil Group	Blend Oil Group	Lard Group	*p*-Value *		
					Time	Group	Time × Group
Age: ≥45 (n = 17–33)							
AST	0	23.52 (19.37, 27.66)	24.21 (21.71, 26.70)	24.11 (21.46, 26.76)	0.002	0.108	0.398
	Δ4	−1.27 (−3.26, 0.72)	−4.00 (−5.89, −2.11)	−2.00 (−4.60, 0.60)			
	Δ8	1.41 (−1.58, 4.41)	−0.68 (−2.47, 1.11)	−0.45 (−3.91, 3.01)			
	Δ12	−0.10 (−2.22, 2.02)	−2.61 (−5.81, 0.59)	0.68 (−5.05, 6.42)			
ALT	0	19.73 (15.70, 23.75)	24.42 (18.21, 30.62)	23.52 (20.40, 26.64)	0.002	0.005	0.034
	Δ4	−0.52 (−2.98, 1.94) ^a^	−5.53 (−9.42, −1.63) ^b^	−2.35 (−4.46, −0.24)			
	Δ8	1.48 (−1.45, 4.42)	−2.32 (−5.43, 0.80)	−1.70 (−4.94, 1.54)			
	Δ12	0.57 (−2.11, 3.24)	−5.72 (−13.82, 2.38)	−1.37 (−7.86, 5.12)			
Age: <45 (n = 58–92)							
AST	0	21.07 (19.87, 22.28)	22.79 (21.51, 24.07)	21.42 (20.02, 22.82)	<0.001	<0.001	0.001
	Δ4	−1.00 (−2.02, 0.02) ^a^	−2.59 (−3.65, −1.53) ^b^	−0.86 (−1.82, 0.10) ^a^			
	Δ8	0.26 (−0.81, 1.33)	−0.76 (−1.89, 0.36)	0.46 (−0.88, 1.80)			
	Δ12	−0.41 (−1.34, 0.53)	−1.77 (−3.05, −0.50) ^b^	1.13 (−0.12, 2.38) ^a^			
ALT	0	20.39 (16.88, 23.89)	23.83 (20.25, 27.40)	21.62 (17.71, 25.52)	0.097	<0.001	<0.001
	Δ4	−0.84 (−3.03, 1.35) ^a^	−3.91 (−6.27, −1.54) ^b^	0.32 (−1.74, 2.37) ^a^			
	Δ8	0.46 (−2.32, 3.23)	−2.49 (−5.06, 0.09)	0.67 (−2.38, 3.72)			
	Δ12	−0.45 (−3.05, 2.14) ^a^	−4.16 (−7.04, −1.28) ^b^	2.72 (0.46, 4.98) ^a^			
Male (n = 27–55)							
AST	0	24.11 (22.24, 25.97)	25.07 (23.28, 26.87)	24.73 (22.59, 26.88)	<0.001	<0.001	0.001
	Δ4	−2.56 (−3.82, −1.30) ^a^	−3.50 (−5.18, −1.82) ^a^	−0.29 (−2.11, 1.54) ^b^			
	Δ8	−0.14 (−1.90, 1.61)	−1.55 (−3.23, 0.13)	0.64 (−2.05, 3.32)			
	Δ12	−1.03 (−2.61, 0.56) ^a^	−3.28 (−5.30, −1.27) ^a^	2.70 (−1.16, 6.56) ^b^			
ALT	0	28.28 (22.86, 33.69)	30.33 (25.21, 35.44)	29.78 (22.99, 36.57)	0.012	<0.001	<0.001
	Δ4	−3.58 (−6.58, −0.58) ^a^	−6.17 (−10.08, −2.25) ^a^	1.51 (−2.55, 5.58) ^b^			
	Δ8	−0.71 (−5.06, 3.63)	−4.75 (−8.65, −0.85)	−0.03 (−6.18, 6.11)			
	Δ12	−2.26 (−6.54, 2.01) ^a^	−9.03 (−14.09, −3.96) ^b^	4.74 (−1.01, 10.49) ^a^			
Female (n = 40–72)							
AST	0	20.17 (18.17, 22.18)	21.30 (20.01, 22.58)	20.54 (19.11, 21.97)	<0.001	0.020	0.076
	Δ4	−0.09 (−1.34, 1.15) ^a^	−2.37 (−3.37, −1.36) ^b^	−1.62 (−2.69, −0.55) ^b^			
	Δ8	1.13 (−0.47, 2.72)	−0.06 (−1.10, 0.97)	0.04 (−1.28, 1.35)			
	Δ12	0.18 (−0.92, 1.28)	−0.71 (−1.99, 0.57)	0.17 (−1.25, 1.59)			
ALT	0	14.70 (12.75, 16.64)	18.20 (15.15, 21.24)	17.68 (15.26, 20.10)	0.139	0.002	0.018
	Δ4	1.17 (−0.69, 3.03) ^a^	−2.67 (−4.54, −0.80) ^b^	−1.38 (−2.56, −0.19) ^b^			
	Δ8	1.75 (−0.32, 3.81)	−0.49 (−2.50, 1.52)	0.25 (−1.58, 2.08)			
	Δ12	1.32 (−0.18, 2.83)	−0.22 (−2.18, 1.74)	0.23 (−1.61, 2.07)			

Values are presented as means (95% CIs). * Comparisons among the three groups were performed using mixed-model analysis of variance; changes from baseline were calculated by subtracting 4-, 8-, and 12-week data from baseline data. AST, aspartate aminotransferase; ALT, alanine aminotransferase; Δ, change from baseline. ^a,b^ Different superscripts indicate significant differences between groups. No superscript means no difference compared with any other group.

**Table 6 foods-12-01894-t006:** Changes of other liver function indicators from baseline and comparisons among the three groups.

Variable	Week	Soybean Oil Group (n = 93–116)	Blend Oil Group (n = 81–116)	Lard Group (n = 79–113)	Reference Intervals	*p*-Value *		
						Time	Group	Time × Group
ALP (U/L)	0	72.59 (68.94, 76.24)	71.60 (68.06, 75.15)	73.68 (70.23, 77.13)	35.0–100.0	<0.001	0.486	0.927
	Δ4	−12.38 (−13.81, −10.96)	−12.57 (−14.53, −10.62)	−12.26 (−14.23, −10.29)				
	Δ8	−2.21 (−3.95, −0.47)	−2.75 (−4.13, −1.37)	−1.37 (−3.26, 0.52)				
	Δ12	−11.49 (−13.08, −9.89)	−11.81 (−13.45, −10.18)	−10.95 (−12.72, −9.18)				
γ-GT (U/L)	0	22.41 (19.55, 25.26)	24.95 (22.06, 27.84)	24.09 (21.31, 26.87)	7.0–45.0	<0.001	0.014	0.091
	Δ4	−1.59 (−3.04, −0.14)	−3.51 (−4.89, −2.13)	−2.44 (−3.96, −0.91)				
	Δ8	−0.41 (−1.84, 1.02)	−0.69 (−1.91, 0.53)	0.28 (−1.12, 1.68)				
	Δ12	−0.59 (−1.61, 0.42) ^a^	−3.15 (−4.72, −1.58) ^b^	−0.66 (−2.84, 1.51) ^a^				
TP (g/L)	0	76.34 (75.66, 77.02)	75.69 (74.96, 76.42)	76.68 (75.92, 77.44)	65.0–85.0	<0.001	0.368	0.762
	Δ4	0.36 (−0.23, 0.96)	0.35 (−0.29, 1.00)	−0.11 (−0.89, 0.66)				
	Δ8	−1.96 (−2.60, −1.32)	−1.44 (−2.07, −0.81)	−1.96 (−2.63, −1.29)				
	Δ12	−3.48 (−4.16, −2.80)	−3.15 (−3.92, −2.38)	−3.36 (−4.18, −2.54)				
ALB (g/L)	0	46.60 (46.12, 47.07)	46.29 (45.88, 46.69)	46.41 (46.01, 46.80)	40.0–55.0	<0.001	0.133	0.593
	Δ4	0.45 (0.10, 0.81)	0.54 (0.19, 0.90)	0.43 (0.04, 0.82)				
	Δ8	−0.57 (−1.60, 0.46)	0.27 (−0.13, 0.68)	0.12 (−0.29, 0.53)				
	Δ12	0.63 (0.23, 1.03)	0.92 (0.48, 1.36)	0.79 (0.37, 1.21)				
GLB (g/L)	0	29.74 (29.14, 30.34)	29.40 (28.77, 30.04)	30.27 (29.61, 30.94)	20.0–40.0	<0.001	0.110	0.333
	Δ4	0.06 (−0.33, 0.46)	−0.19 (−0.57, 0.18)	−0.49 (−0.95, −0.03)				
	Δ8	−1.85 (−2.20, −1.50)	−1.71 (−2.07, −1.36)	−2.15 (−2.51, −1.80)				
	Δ12	−4.03 (−4.42, −3.63)	−4.07 (−4.49, −3.66)	−4.15 (−4.62, −3.67)				
A/G (%)	0	1.59 (1.55, 1.63)	1.60 (1.56, 1.64)	1.55 (1.52, 1.59)	1.2–2.4	<0.001	0.310	0.402
	Δ4	0.02 (0.00, 0.03)	0.03 (0.01, 0.05)	0.04 (0.02, 0.05)				
	Δ8	0.11 (0.09, 0.13)	0.11 (0.09, 0.13)	0.13 (0.11, 0.15)				
	Δ12	0.28 (0.26, 0.31)	0.30 (0.27, 0.32)	0.28 (0.25, 0.30)				
T-BIL (μmol/L)	0	14.36 (13.37, 15.34)	13.95 (12.98, 14.93)	14.33 (13.24, 15.41)	2.0–20.0	0.002	0.978	0.929
	Δ4	0.81 (−0.10, 1.71)	0.85 (−0.22, 1.91)	0.72 (−0.08, 1.52)				
	Δ8	0.24 (−0.63, 1.11)	−0.02 (−1.02, 0.98)	−0.47 (−1.50, 0.56)				
	Δ12	0.42 (−0.51, 1.35)	0.68 (−0.25, 1.61)	1.02 (−0.03, 2.07)				
D-BIL (μmol/L)	0	3.67 (3.40, 3.94)	3.63 (3.36, 3.91)	3.61 (3.32, 3.90)	0.0–6.0	<0.001	0.309	0.286
	Δ4	−0.09 (−0.32, 0.15)	0.04 (−0.29, 0.36)	−0.12 (−0.38, 0.13)				
	Δ8	1.27 (1.02, 1.52) ^a^	1.17 (0.87, 1.47)	0.79 (0.49, 1.10) ^b^				
	Δ12	−0.24 (−0.47, 0.00)	−0.16 (−0.41, 0.08)	−0.11 (−0.40, 0.17)				
I-BIL (μmol/L)	0	10.69 (9.95, 11.43)	10.37 (9.64, 11.10)	10.71 (9.90, 11.53)	0.0–13.6	<0.001	0.900	0.954
	Δ4	0.89 (0.20, 1.58)	0.94 (0.14, 1.74)	0.90 (0.31, 1.50)				
	Δ8	−1.03 (−1.71, −0.34)	−1.19 (−1.93, −0.45)	−1.26 (−2.03, −0.49)				
	Δ12	0.68 (−0.06, 1.41)	0.84 (0.14, 1.54)	1.27 (0.49, 2.06)				

All values are means (95% CIs). * Comparisons among the three groups were performed using mixed-model analysis of variance; changes from baseline were calculated by subtracting 4-, 8-, and 12-week data from baseline data. ALP, alkaline phosphatase; γ-GT, gamma-glutamyltransferase; TP, total protein; ALB, albumin; GLB, globulin; A/G, albumin/globulin; T-BIL, total bilirubin; D-BIL, direct bilirubin; I-BIL, indirect bilirubin; Δ, change from baseline. ^a,b^ Different superscripts indicate significant differences between groups. No superscript means no difference compared with any other group.

**Table 7 foods-12-01894-t007:** Changes of cardiometabolic risk factors from baseline and comparisons among the three groups.

Variable	Week	Soybean Oil Group (n = 93–116)	Blend Oil Group (n = 81–116)	Lard Group (n = 79–113)	*p*-Value *		
					Time	Group	Time × Group
Weight (kg)	0	65.0 (62.5, 67.5)	67.0 (64.5, 69.5)	64.4 (62.1, 66.6)	<0.001	0.399	0.889
	Δ4	−0.38 (−0.61, −0.16)	−0.23 (−0.51, 0.06)	−0.25 (−0.59, 0.08)			
	Δ8	−0.27 (−0.56, 0.03)	0.02 (−0.32, 0.36)	−0.14 (−0.54, 0.25)			
	Δ12	−0.44 (−0.79, −0.10)	−0.32 (−0.67, 0.04)	−0.33 (−0.84, 0.18)			
WC (cm)	0	78.2 (76.0, 80.3)	78.7 (76.5, 81.0)	77.1 (75.0, 79.2)	<0.001	0.749	0.882
	Δ4	3.42 (2.43, 4.41)	3.33 (2.62, 4.04)	3.17 (2.53, 3.82)			
	Δ8	1.60 (0.78, 2.42)	2.32 (1.51, 3.13)	1.64 (0.87, 2.41)			
	Δ12	−0.59 (−1.29, 0.11)	−0.51 (−1.46, 0.44)	−0.32 (−1.25, 0.60)			
Glucose (mmol/L)	0	4.73 (4.60, 4.87)	4.76 (4.62, 4.90)	4.84 (4.64, 5.04)	<0.001	0.473	0.818
	Δ4	−0.09 (−0.20, 0.02)	−0.09 (−0.21, 0.02)	−0.15 (−0.25, −0.04)			
	Δ8	−0.21 (−0.35, −0.07)	−0.27 (−0.39, −0.15)	−0.20 (−0.35, −0.06)			
	Δ12	−0.26 (−0.42, −0.11)	−0.39 (−0.52, −0.26)	−0.35 (−0.48, −0.23)			
SBP (mm Hg)	0	115.8 (113.5, 118.0)	117.4 (114.9, 119.9)	115.6 (113.8, 117.4)	<0.001	0.001	0.029
	Δ4	−2.13 (−4.27, 0.01)	−3.68 (−5.62, −1.74)	−1.34 (−3.47, 0.79)			
	Δ8	−3.46 (−5.93, −0.99)	−6.71 (−9.33, −4.08) ^b^	−2.91 (−5.27, −0.55) ^a^			
	Δ12	−3.22 (−5.58, −0.86)	−5.83 (−8.39, −3.26) ^b^	−1.25 (−3.81, 1.32) ^a^			
DBP (mm Hg)	0	74.9 (73.3, 76.5)	76.0 (74.1, 77.9)	74.2 (72.6, 75.7)	<0.001	<0.001	0.003
	Δ4	−1.65 (−3.39, 0.09) ^a^	−4.26 (−5.99, −2.52) ^b^	−0.26 (−2.36, 1.84) ^a^			
	Δ8	−2.34 (−3.83, −0.85)	−4.44 (−6.22, −2.65) ^b^	−0.91 (−2.89, 1.07) ^a^			
	Δ12	1.54 (−0.87, 3.94)	0.54 (−1.85, 2.94)	3.44 (1.05, 5.84)			
TC (mmol/L)	0	4.56 (4.43, 4.70)	4.70 (4.56, 4.85)	4.66 (4.51, 4.81)	<0.001	0.460	0.798
	Δ4	−0.34 (−0.42, −0.26)	−0.35 (−0.43, −0.26)	−0.37 (−0.45, −0.28)			
	Δ8	−0.25 (−0.36, −0.15)	−0.18 (−0.27, −0.10)	−0.17 (−0.27, −0.06)			
	Δ12	−0.23 (−0.33, −0.13)	−0.20 (−0.30, −0.10)	−0.15 (−0.27, −0.03)			
LDL-C (mmol/L)	0	2.55 (2.44, 2.66)	2.67 (2.54, 2.79)	2.62 (2.50, 2.73)	0.206	0.674	0.912
	Δ4	−0.01 (−0.07, 0.04)	−0.02 (−0.09, 0.05)	−0.03 (−0.10, 0.03)			
	Δ8	−0.04 (−0.12, 0.04)	0.00 (−0.09, 0.10)	0.00 (−0.07, 0.08)			
	Δ12	−0.06 (−0.13, 0.01)	−0.05 (−0.12, 0.03)	0.00 (−0.09, 0.08)			
TG (mmol/L)	0	1.12 (0.97, 1.26)	1.09 (0.99, 1.20)	1.13 (0.99, 1.27)	0.001	0.606	0.961
	Δ4	0.02 (−0.07, 0.11)	0.02 (−0.07, 0.11)	0.04 (−0.07, 0.16)			
	Δ8	0.03 (−0.07, 0.13)	−0.02 (−0.10, 0.07)	0.04 (−0.04, 0.12)			
	Δ12	−0.13 (−0.22, −0.04)	−0.12 (−0.22, −0.03)	−0.08 (−0.21, 0.04)			
HDL-C (mmol/L)	0	1.68 (1.61, 1.75)	1.70 (1.65, 1.75)	1.69 (1.64, 1.75)	<0.001	0.397	0.680
	Δ4	−0.33 (−0.36, −0.30)	−0.33 (−0.36, −0.30)	−0.34 (−0.37, −0.30)			
	Δ8	−0.30 (−0.33, −0.26)	−0.28 (−0.31, −0.24)	−0.26 (−0.29, −0.23)			
	Δ12	−0.22 (−0.25, −0.18)	−0.18 (−0.22, −0.15)	−0.20 (−0.24, −0.17)			
TC/HDL-C	0	2.81 (2.70, 2.93)	2.82 (2.72, 2.93)	2.81 (2.71, 2.91)	<0.001	0.591	0.830
	Δ4	0.41 (0.36, 0.46)	0.42 (0.36, 0.48)	0.41 (0.34, 0.48)			
	Δ8	0.37 (0.31, 0.44)	0.38 (0.31, 0.45)	0.41 (0.33, 0.49)			
	Δ12	0.23 (0.17, 0.28)	0.18 (0.12, 0.24)	0.24 (0.18, 0.31)			
FFA (μmol/L)	0	754.3 (703.7, 804.9)	730.4 (678.8, 781.9)	778.7 (729.8, 827.6)	<0.001	0.564	0.913
	Δ4	0.4 (−62.9, 63.7)	9.3 (−52.4, 70.9)	−22.0 (−82.7, 38.7)			
	Δ8	−129.4 (−184.4, −74.4)	−114.8 (−175.7, −53.8)	−148.2 (−204.9, −91.6)			
	Δ12	−127.5 (−179.3, −75.8)	−117.2 (−169.8, −64.7)	−101.5 (−158.5, −44.5)			

All values are means (95% CIs). * Comparisons among the three groups were performed using mixed-model analysis of variance; changes from baseline were calculated by subtracting 4-, 8-, and 12-week data from baseline data. WC, waist circumference; SBP, systolic blood pressure; DBP, diastolic blood pressure; FFA, free fatty acid; HDL-C, high-density lipoprotein cholesterol; LDL-C, low-density lipoprotein cholesterol; TC, total cholesterol; TG, triglyceride; Δ, change from baseline. ^a,b^ Different superscripts indicate significant differences between groups. No superscript means no difference compared with any other group.

**Table 8 foods-12-01894-t008:** Changes of fatty acid composition of erythrocyte membrane in subjects from baseline and comparisons among the three groups.

Fatty Acids	Week	Soybean Oil Group (n = 84)	Blend Oil Group (n = 76)	Lard Group (n = 83)	*p*-Value *		
					Time	Group	Time × Group
C14:0 (myristic acid)	0	0.14 (0.04)	0.13 (0.04)	0.13 (0.04)	0.004	0.178	0.178
	Δ8	0.00 (0.07)	0.02 (0.06)	0.02 (0.07)			
C16:0 (palmitic acid)	0	18.08 (1.98)	17.94 (1.89)	17.81 (1.18)	<0.001	0.178	0.178
	Δ8	0.43 (2.59)	0.78 (2.27)	1.07 (1.79)			
C18:0 (stearic acid)	0	10.71 (1.15)	10.60 (1.32)	10.64 (0.97)	<0.001	0.664	0.664
	Δ8	−0.58 (1.25)	−0.48 (1.20)	−0.42 (1.14)			
C22:0 (behenic acid)	0	0.43 (0.21)	0.39 (0.20)	0.50 (0.35)	<0.001	0.053	0.053
	Δ8	−0.13 (0.23)	−0.11 (0.22) ^a^	−0.21 (0.37) ^b^			
C24:0 (lignoceric acid)	0	2.56 (0.59)	2.56 (0.50)	2.51 (0.51)	<0.001	0.731	0.731
	Δ8	−0.90 (0.58)	−0.89 (0.62)	−0.83 (0.60)			
C16:1n-7 (palmitoleic acid)	0	0.27 (0.08)	0.28 (0.09)	0.28 (0.09)	0.197	0.863	0.863
	Δ8	0.00 (0.10)	−0.01 (0.10)	−0.01 (0.11)			
C18:1n-9 (oleic acid)	0	14.21 (1.49)	13.96 (1.49)	13.94 (1.17)	<0.001	0.270	0.270
	Δ8	1.28 (1.67)	1.32 (1.75)	1.64 (1.31)			
C22:1n-9 (erucic acid)	0	0.85 (0.43)	0.80 (0.43)	0.92 (0.53)	<0.001	0.760	0.760
	Δ8	−0.34 (0.48)	−0.37 (0.52)	−0.40 (0.53)			
C18:2n-6 (linoleic acid)	0	16.20 (1.43)	16.34 (1.60)	16.20 (1.75)	0.010	0.118	0.118
	Δ8	−0.01 (1.19) ^a^	−0.21 (1.31)	−0.44 (1.44) ^b^			
C20:3n-6 (dihomo-γ-linolenic acid)	0	1.32 (0.32)	1.45 (0.34)	1.35 (0.33)	<0.001	0.118	0.118
	Δ8	−0.35 (0.29)	−0.42 (0.35) ^a^	−0.33 (0.29) ^b^			
C20:4n-6 (arachidonic acid)	0	23.20 (2.71)	23.51 (2.41)	23.29 (2.00)	<0.001	0.989	0.989
	Δ8	0.62 (2.98)	0.57 (2.38)	0.57 (2.03)			
C18:3n-3 (linolenic acid)	0	0.20 (0.11)	0.20 (0.11)	0.25 (0.16)	0.082	0.001	0.001
	Δ8	0.06 (0.14) a	0.02 (0.11) a	−0.03 (0.17) b			
C20:5n-3 (eicosapentaenoic acid)	0	0.39 (0.12)	0.42 (0.13)	0.45 (0.16)	<0.001	0.119	0.119
	Δ8	−0.04 (0.09)	−0.03 (0.11)	−0.07 (0.14)			
C22:6n-3 (docosahexaenoic acid)	0	8.17 (1.46)	8.28 (1.39)	8.46 (1.43)	0.563	0.141	0.141
	Δ8	0.25 (1.41) ^a^	0.05 (1.31)	−0.15 (1.18) ^b^			

Values are presented as means (SD). Δ, change from baseline. * Comparisons among the three groups were performed using mixed-model analysis of variance; changes from baseline were calculated by subtracting 8-week data from baseline data. ^a,b^ Different superscripts indicate significant differences between groups. No superscript means no difference compared with any other group.

## Data Availability

The data presented in this study are available upon request from the corresponding authors.

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
