# Peer review of "Effect of Lard or Plus Soybean Oil on Markers of Liver Function in Healthy Subjects: A Randomized Controlled-Feeding Trial"

_foods, 2023, doi:10.3390/foods12091894_

Round 1

Reviewer 1 Report

To address the suggestions made in the document in order to improve its quality.

Author Response

1. Comentado [E1]: I suggest including in the paragraph the importance of liver function tests for lard and soybean oil consumption, highlighting AST and ALT markers.

Response:  As suggested, the following information has been added to the manuscript to state the importance of liver function tests for lard and soybean oil consumption, highlighting AST and ALT markers (P2, L61-63):

Recently, several animal studies demonstrated that lard plus soybean oil alleviated NAFLD compared with soybean oil partly by reducing the AST and ALT [4] and had a remarkable anti-obesity effect compared with lard and soybean oil alone [19].

Note: we have addressed the other points made by Reviewer 1 in the manuscript.

Thank you very much for your attention and look forward to hearing from you soon.

Sincerely yours,

Duo Li,

FRCS, FIUNS, PhD

Chief Professor of Nutrition

Institute of Nutrition & Health, Qingdao University, China

Reviewer 2 Report

Dear authors

The article titled “Effect of Lard or Plus Soybean Oil on Markers of Liver Function in Healthy Subjects: A Randomized Controlled-Feeding 3 Trial” is interesting and highlights the significance of using Lard and its blend with soybean oil to reduce the chances of non-alcoholic fatty liver disease in healthy individuals. It is a well organized and well-written article. However, the authors must consider the following points to discuss in the manuscript to make the data clearer and more understandable.

1.         Authors used 30g of oil/day during this study. Does this the average daily consumption of oil by a normal human? What was the rationale behind choosing this amount? Please mention it in the methods section in one sentence.

2.         What was the percentage of male and female participants who completed the study? Table 3 does not have any information about male participants.

3.         Although this study shows the specific blend of oil is hepatoprotective, it could cause cardiovascular problems. So, it would improve the discussion if the authors shed light on the consequences of a large intake of oil/day and provide some suggestions on how to avoid other fat-intake-related health issues.

4.         Authors have discussed the limitations and strengths of this study but add future directions and suggestions which must be considered and included in future studies to produce more conclusive and widely applicable data.

Reviewer 3 Report

The paper is very interesting and well sound.

English language is fine.

Statistics seem appropriate. The number of subjects as well.

The references are updated.

I have no major concerns.

The single minor concern is the following:

in the discussion a paragraph dealing with the meaning of a tiny altough significant change in AST/ALT in health improvement of humans should be inserted.

Apart this requested addition I have to add nothing.

Author Response

1. The single minor concern is the following: in the discussion a paragraph dealing with the meaning of a tiny although significant change in AST/ALT in health improvement of humans should be inserted.

Response: Thank you for your advice. As suggested, we have added the following information to the manuscript (P13-14, L333-338):

ALT is primarily localized to the liver but AST is present in a wide variety of tissues like the heart, skeletal muscle, kidney, brain and liver [24]. Both AST and ALT levels are increased to some extent in almost all liver diseases due to hepatocellular necrosis, including NAFLD, cirrhosis, non-alcoholic steatohepatitis (NASH), etc [24]. In the present study, the significant reductions in both AST and ALT in the blend oil group suggested that blend oil is likely to decrease hepatocellular necrosis, which might be beneficial to prevent liver diseases, including NAFLD, which affected approximately 30% of the population [25].

Thank you very much for your attention and look forward to hearing from you soon.

Sincerely yours,

Duo Li,

FRCS, FIUNS, PhD

Chief Professor of Nutrition

Institute of Nutrition & Health, Qingdao University, China